# Epidemiological characteristics and prevalence rates of research reproducibility across disciplines: A scoping review of articles published in 2018-2019

Kelly D Cobey[1,2]*, Christophe A Fehlmann[2,3], Marina Christ Franco[4,5], Ana Patricia Ayala[6], Lindsey Sikora[7], Danielle B Rice[4,8], Chenchen Xu[4,9], John PA Ioannidis[10], Manoj M Lalu[4,11,12], Alixe Ménard[4], Andrew Neitzel[4,9], Bea Nguyen[4,9], Nino Tsertsvadze[4], David Moher[2,4]

[1]Heart Institute, University of Ottawa, Ottawa, Canada; [2]School of Epidemiology and Public Health, University of Ottawa, Ottawa, Canada; [3]Department of Anaesthesiology, Clinical Pharmacology, Intensive Care and Emergency Medicine, Geneva University Hospitals, Geneva, Switzerland; [4]Centre for Journalology, Clinical Epidemiology Program, Ottawa Hospital Research Institute, Ottawa, Canada; [5]School of Dentistry, Federal University of Pelotas, Pelotas, Brazil; [6]Gerstein Science Information Centre, University of Toronto, Toronto, Canada; [7]Health Sciences Library, University of Ottawa, Ottawa, Canada; [8]Department of Psychology, McGill University, Montreal, Canada; [9]Department of Medicine, University of Ottawa, Ottawa, Canada; [10]Departments of Medicine, of Epidemiology and Population Health, of Biomedical Data Science, and of Statistics, and Meta-Research Innovation Center at Stanford, Stanford University, Stanford, United States; [11]Department of Anesthesiology and Pain Medicine, University of Ottawa, Ottawa, Canada; [12]Regenerative Medicine Program, Ottawa Hospital, Ottawa, Canada

*For correspondence: kcobey@ottawaheart.ca

Competing interest: The authors declare that no competing interests exist.

## Abstract

**Background:** Reproducibility is a central tenant of research. We aimed to synthesize the literature on reproducibility and describe its epidemiological characteristics, including how reproducibility is defined and assessed. We also aimed to determine and compare estimates for reproducibility across different fields.

**Methods:** We conducted a scoping review to identify English language replication studies published between 2018 and 2019 in economics, education, psychology, health sciences, and biomedicine. We searched Medline, Embase, PsycINFO, Cumulative Index of Nursing and Allied Health Literature – CINAHL, Education Source via EBSCOHost, ERIC, EconPapers, International Bibliography of the Social Sciences (IBSS), and EconLit. Documents retrieved were screened in duplicate against our inclusion criteria. We extracted year of publication, number of authors, country of affiliation of the corresponding author, and whether the study was funded. For the individual replication studies, we recorded whether a registered protocol for the replication study was used, whether there was contact between the reproducing team and the original authors, what study design was used, and what the primary outcome was. Finally, we recorded how reproducibilty was defined by the authors, and whether the assessed study(ies) successfully reproduced based

on this definition. Extraction was done by a single reviewer and quality controlled by a second reviewer.

**Results:** Our search identified 11,224 unique documents, of which 47 were included in this review. Most studies were related to either psychology (48.6%) or health sciences (23.7%). Among these 47 documents, 36 described a single reproducibility study while the remaining 11 reported at least two reproducibility studies in the same paper. Less than the half of the studies referred to a registered protocol. There was variability in the definitions of reproduciblity success. In total, across the 47 documents 177 studies were reported. Based on the definition used by the author of each study, 95 of 177 (53.7%) studies reproduced.

**Conclusions:** This study gives an overview of research across five disciplines that explicitly set out to reproduce previous research. Such reproducibility studies are extremely scarce, the definition of a successfully reproduced study is ambiguous, and the reproducibility rate is overall modest.

**Funding:** No external funding was received for this work

## Editor's evaluation

It has been recognized since the beginning of science that science can always be made more rigorous. Indeed, it is part of the ethos and very nature of the scientific method and the scientific attitude, as Lee McIntyre describes in his brilliant book by that title, to be constantly striving for improvements in rigor. Yet, we know that there are breaches in rigor, reproducibility, and transparency of research conduct and reporting. Such breaches have been highlighted more intensively, or at least so it seems, for more than a decade. The field recognizes that we need to go beyond platitudinous recognition that there is always opportunity for improvement in rigor and that such improvements are vital, to identifying those key leverage points where efforts can have the most positive near-term effects. Identifying domains in which reproducibility is greater or lesser than in other domains can aid in that regard. Thus, this article represents a constructive step in identifying key opportunities for bettering our science and that is something that every scientist can stand behind.

## Introduction

Reproducibility is a central tenant of research. Reproducing previously published studies helps us to discern discoveries from false leads. The lexicon around reproducibility studies is diverse and poorly defined (*Goodman et al., 2016*). Here, we loosely use Nosek and Errington's definition: '*a study for which any outcome would be considered diagnostic evidence about a claim from prior research*' (*Nosek and Errington, 2020a*). Most scientific studies are never formally reproduced and some disciplines have lower rates of reproducibility attempts than others. For example, in education research, an analysis published in 2014 of all publications in the discipline's top 100 journals found that only 0.13% (221 out of 164,589) of the published articles described an independent reproducibility study (*Makel and Plucker, 2014*). There is rising concern about the reproducibility of research and increasing interest in enhancing research transparency (e.g. *Buck, 2015*; *Munafò et al., 2017*; *Collins and Tabak, 2014*; *Begley and Ioannidis, 2015*).

Knowledge about rates of reproducibility is currently dominated by a handful of well-known projects examining the reproducibility of a group of studies within a field. For example, a project estimating the reproducibility of 100 psychology studies published in three leading journals found that just 36% had statistically significant results, compared to 97% of the original studies. Just 47% of the studies had effect sizes that were within the bounds of the 95% confidence interval of the original study (*Open Science Collaboration, 2015*). Estimates of reproducibility in economics are similarly modest or low: a large-scale study attempting to reproduce 67 papers was only able to reproduce 22 (33%) of these (*Chang and Li, 2015*). This same study showed that when teams attempting to reproduce research involved one of the original study authors as a co-author on the project, rates of reproducibility increased. This may suggest that detailed familiarity with the original study method increases the likelihood of reproducing the research findings. A cancer biology reproducibility project launched in 2013 to independently reproduce several high-profile papers has produced rather sobering results a decade later. Most of the selected studies could not even be attempted to be reproduced (e.g.

it was not clear what had been done originally) and among those that an attempt to reproduce was made, most did not seem to produce consistent results (although the exact reproducibility rate depends on the definition of reproducibility) and the effect of the reproduced studies was only 15% of the original effect (*Errington et al., 2021a*; *Kane and Kimmelman, 2021*; *Errington et al., 2021b*).

In medicine, studies that do not reproduce in clinic may exaggerate patient benefits and harms (*Le Noury et al., 2015*) especially when clinical decisions are based on a single study. Despite this and other potential consequences we know very little about what predicts research reproducibility. No data exists which provides systematic estimates for reproducibility across multiple disciplines or addresses why disciplines might vary in their rates of reproducibility. Failure to empirically examine reproducibility is regrettable: without research we can't identify actions to take that could drive improvements in research reproducibility. This contributes to research waste (*Nasser et al., 2017*; *Ioannidis et al., 2014*; *Freedman et al., 2015*).

We set out to broadly examine the reproducibility literature across five disciplines and report on characteristics including how reproducibility is defined, assessed, and document prevalence rates for reproducibility. We focused on studies that explicitly described themselves as reproducibility or replication studies addressing reproductions of previously published work.

## Methods

Our study was conducted using the framework proposed by *Arksey and O'Malley, 2005* and the related update by *Levac et al., 2010*, and follows a five stage process: (1) identifying the research question, (2) identifying relevant studies, (3) study selection, (4) charting the data, (5) collating, summarizing and reporting the results.

### Protocol registration and Open Science statement

This protocol was shared on the Open Science Framework prior to initiating the study (https://osf.io/59nw4/). We used the PRISMA-ScR (*Tricco et al., 2018*) checklist to guide our reporting. Study data and materials are also available on the Open Science Framework (https://osf.io/wn7gm/).

### Eligibility criteria

#### Inclusion criteria

We included all quantitative reproducibility studies within the fields of economics, education, psychology, health sciences and biomedicine that were published in 2018 or 2019. Definitions we established for each discipline can be found in Appendix 1. We included all studies that explicitly self-described as a replication or a reproducibility study in which a previously published quantitative study is referred to and conducted again. As per Nosek and Errington's definition (*Nosek and Errington, 2020a*), we did not require that methods be perfectly matched between the original study and the replication if the author described the study as a replication. We excluded studies where the main intention of the work was not framed as a reproducibility project.

#### Exclusion criteria

We excluded complementary and alternative forms of medicine as defined by the National Institutes of Health's National Center for Complementary and Integrative Health (https://www.nccih.nih.gov/) for feasibility based on pilot searches. We excluded literature that was not published in English for feasibility, or that described exclusively qualitative research. We excluded conference proceedings, commentaries, narrative reviews, systematic reviews (not original research), and clinical case studies. We also excluded studies that described a replication of a study but where the original study was reporting in the same publication.

### Information sources and search strategy

Our search strategy was developed by trained information specialists (APA, LS), and peer reviewed using the PRESS guideline (*McGowan et al., 2016*). We restricted our search to the years 2018 and 2019 in order to maintain feasibility of this study given our available resources for screening and data extraction. We searched the following databases: Medline via Ovid (1946–2020), Embase via Ovid (1947–2020), PsycINFO via Ovid (1806–2020), Cumulative Index of Nursing and Allied Health

Literature – CINAHL (1937–2020), Education Source via EBSCOHost (1929–2020), ERIC via Ovid (1966–2020), EconPapers (inception – 2020), International Bibliography of the Social Sciences (IBSS) via ProQuest (1951–2020), and EconLit via EBSCOHost (1969–2020). We performed forward and backward citation analysis of articles included for data extraction in Scopus and Web of Science (platforms including Science Citation Index Expanded (SCI-EXPANDED) -–1900-present and Social Sciences Citation Index (SSCI) -–1900-present) to identify additional potential documents for inclusion. All searches are reported using PRISMA-S (*Rethlefsen et al., 2021*). For full search details please see Appendix 2. A related supplementary search was developed a priori in which we searched preprint servers and conducted forward and backward citation searching (Appendix 3).

## Selection of sources of evidence and data charting process

Search results from the databases were imported into *Distiller SR, 2023* (Evidence Partners, Ottawa, Canada) and de-duplicated. Search results from the supplementary searching were uploaded into Endnote, de-duplicated, and then uploaded into DistillerSR for screening. Team members involved in study screening (KDC, CAF, MCF, DBR, CX, AN, BN, LS, NT, APA) initially screened the titles and abstracts of 50 records and then reviewed conflicts to ensure high level of agreement among screeners (>90%). After piloting was complete, all potentially relevant documents were screened in duplicate using the liberal accelerated method in which records move to full-text screening if one or more reviewers indicate unclear or yes with regards to potential inclusion and two reviewers were required to exclude a record. Then, all included documents were screened in duplicate to ensure they met all eligibility requirements. All conflicts were resolved by consensus or, when necessary, third-party arbitration (MML, DM). The study screening form can be found in Appendix 4 and 5.

## Data extraction

Two team members (MCF, CAF) extracted document characteristics. Prior to extraction a series of iterative pilot tests were done on included documents to ensure consistency between extractors. We extracted information including publication year, funding information (if funded, funder type), number of authors, ethics approval, study design, and open science practices (study registration, data sharing) from each included document. We also categorized each included documents based on its discipline area (e.g. Economics, Education, Psychology, Health Sciences, Biomedicine, or any combination of these fields) and whether a single original study was being reproduced or if the paper reported the results from reproducing more than one original study. When a single study was reproduced more than once (e.g. different labs all replicated one study) we classified this as a 'single' reproducibility study. We extracted what the stated primary outcome was. If there was no primary outcome stated, we recorded this, and extracted the first stated outcome described in each document. Finally, we extracted what the results of the reproducibility project as reported by the authors (replicated, not replicated, mixed finding) and categorized method by which the authors of each relevant document assess reproducibility (e.g. comparison of effect sizes, statistical significance from p-values). Where relevant we extracted p-values and related statistical information. This allowed us to test the proportion of reproducible results that were statistically significant. The study extraction form can be found in Appendix 6. In instances where documents described multiple sub-studies, we recorded this and then extracted information from all unique quantitative studies describing a reproducibility study.

## Piloting

Team members extracting data (CAF, MCF) performed a calibration pilot test prior to the onset of full-text screening and extraction. Specifically, a series of included documents were then extracted independently. The team then met to discuss differences in extraction between team members and challenges encountered before extracting from subsequent documents. This was repeated until consensus was reached. Extraction was then done by a single reviewer with a second reviewer doing quality control for all documents. Conflicts were resolved by consensus or, when necessary, third-party arbitration (KDC, DBR, MML, DM).

## Synthesis of results

SPSS 27 (Microsoft) (*IBM Support, 2023*) was used for data analysis. The characteristics of all included documents are presented using frequencies and percentages and described narratively. We report

descriptive statistics where relevant. We then report frequency characteristics of the reproducibility studies, and which reproduced based on authors description of their findings (i.e. using the varied definitions of replication that exist in the literature), per discipline. Next, we describe how factors such as team size, team composition, and discipline relate to the reproducibility study. We compared these factors based on how authors defined reproducibility as well as based on the definition that results were statistically significant (at a conventional threshold of p<0.05).

Text-based responses (e.g. primary outcome) underwent content analysis and are described in thematic groups using frequencies and percentages. All content analysis was done by two independent investigators (CF, KDC) using Microsoft Excel.

## Results
### Open science
Data and materials are available on the Open Science Framework (https://osf.io/wn7gm/).

### Protocol amendments
In our protocol, we specified we would include all documents that explicitly self-describe themselves as a reproducibility or replication study. Over the course of the scoping review we encountered studies that described themselves as being a replication or reproducibility study, but in fact did not describe work that met this definition (e.g. a longitudinal study reporting a new cross sectional report of the data; a study with the goal of replicating a concept rather than a specific study). In these instances, we excluded the study despite the authors description that it was a reproducibility study. Studies describing themselves as being a replication but that explicitly specified that they used a novel method were also excluded, as they did not set out with the explicit goal of replicating previous research approach. We encountered a study that was included where the results were arranged by outcomes but not studies being replicated, in this instance we were unable to determine how the results corresponded to the studies the author listed they reproduced. To accommodate this, we modified our extraction form to include an item indicating that extraction of sub-study information was unclear and could not be performed. We had indicated we would record whether the research involved the study of humans or animals, and if so, how many. We did not include these items on the extraction form after piloting as we found reporting of N to be incomplete making accuracy challenging. We also do not present a re-analysis of the reproducibility studies where we recalculate the rate at which studies reproduce comparatively by discipline given the relatively small representation of disciplines outside of psychology and health science which accounted for 128 (72.3%) of the total studies.

### Selection of sources of evidence
The original search included 16,135 records. An additional 159 novel records were retrieved via grey literature searching: 7 documents were retrieved from searching citations of included documents, 49 were included from searching preprints servers, and 103 were included from citation searching. After de-duplication we screened a total of 11,224 documents, of which, 178 were sought for full-text screening. After full-text screening, 47 documents were included in the review. The remaining 131 documents were excluded because of one or more of these factors: they were not written in English (N=2), we could not obtain a full-text document via our library (N=11), the document was not published in 2018 or 2019 (N=39), the document did not describe an original quantitative research study (N=32), the study was not a quantitative reproduciblity study (N=46), or did not fit in a discipline of interest (N=1). See *Figure 1* for the study flow diagram.

### Characteristics of sources of evidence
The characteristics of the included documents are summarized in *Table 1*. The 47 documents included described a total of 177 reproducibility studies. Thirty-six documents (76.6%) described a single reproducibility study, while 11 (23.4%) documents described multiple reproducibility efforts of distinct studies in a single paper. Twenty-eight (59.6%) documents were published in 2018 while 19 (40.4%) were published in 2019. The corresponding author on most of the documents was based at an institution in the USA 27 (57.4%). The included documents had a median of 3 authors, but papers ranged from having between 1 and 172 authors.

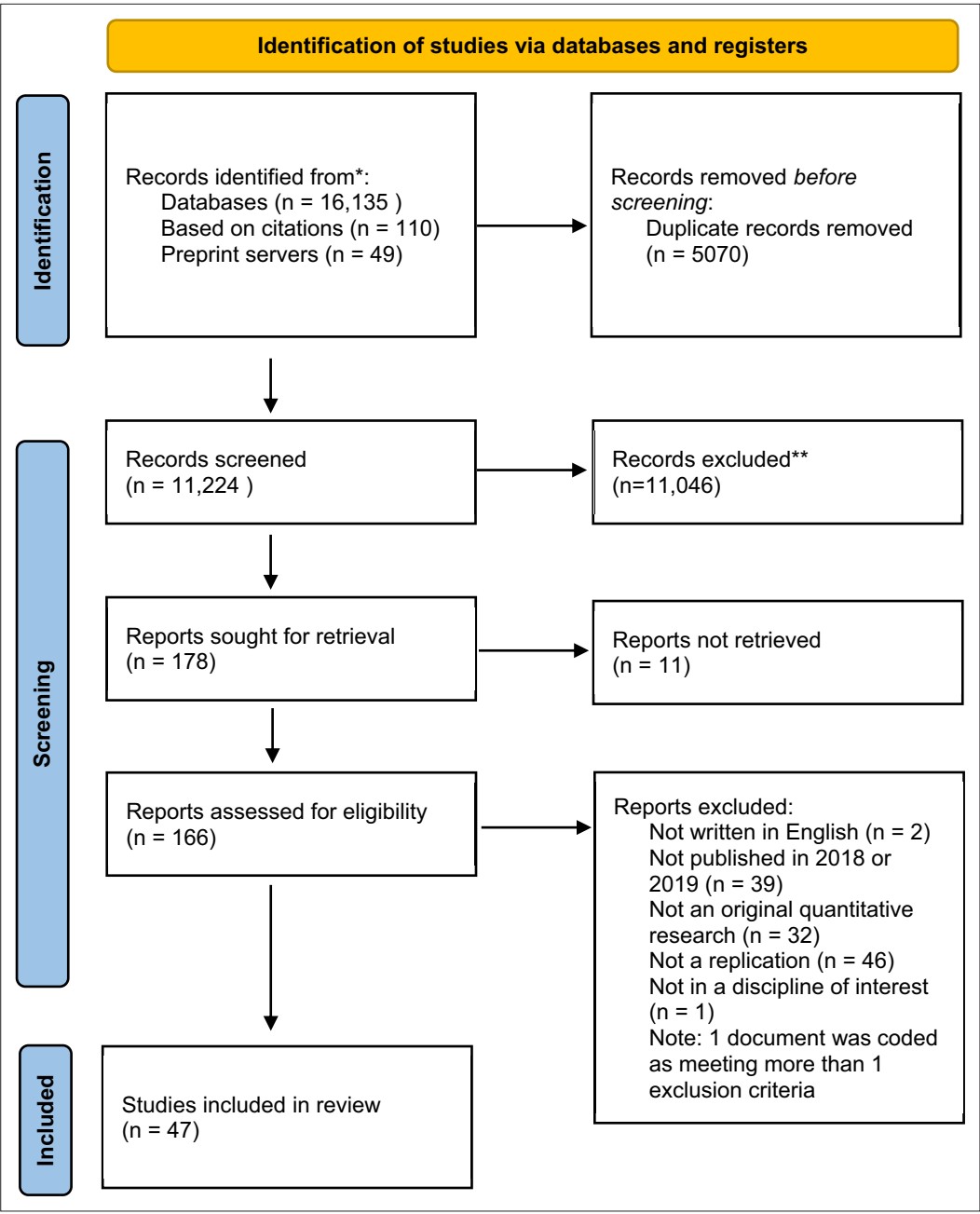

**Figure 1.** Flow diagram of articles.

Thirty-two (68.1%) documents indicated that they received funding, 6 (12.8%) indicated the work was unfunded, and 9 (19.1%) failed to report information about funding. Among documents reporting funding, federal governments were the primary source (N=19, 59.4%). Twenty-three (48.9%) studies reported receiving ethical approval, 10 (21.3%) studies did not report ethical approval, and ethical approval was not relevant for 14 (29.8%) studies.

## Synthesis of reproducibility studies

Most replication studies captured in our sample were in the discipline of psychology (86, 48.6%), followed by health science (42, 23.7%), or were an intersection of our included disciplines (33, 18.6%). There were a relatively smaller number of studies in economics (5, 2.8%), education (5, 2.8%), and biomedicine (6, 3.4%). The most common study designs observed were observational studies (85,

**Table 1.** Characteristics of included documents.

| Characteristic | Categories | All studies | Single replication papers (N=36) | Multiple replication papers (N=11) |
|---|---|---|---|---|
| | | N (%) (unless otherwise indicated) | | |
| What discipline does the work best fit in?* | Biomedicine | 6 (3.4) | 6 (16.7) | - |
| | Economics | 5 (2.8) | 5 (13.9) | - |
| | Education | 5 (2.8) | 1 (2.8) | 1 (9.1) |
| | Health sciences | 42 (23.7) | 9 (25.0) | 4 (36.4) |
| | Psychology | 86 (48.6) | 15 (41.7) | 4 (36.4) |
| | Other (mixture of two or more of the abov1e) | 33 (18.6) | - | 2 (18.2) |
| Year of publication | 2018 | 28 (59.6) | 21 (58.3) | 7 (63.6) |
| | 2019 | 19 (40.4) | 15 (41.7) | 4 (36.4) |
| Country of corresponding author (reported based on Top 3 overall) | USA | 27 (57.4) | 19 (52.8) | 8 (72.7) |
| | The Netherlands | 4 (8.5) | 3 (8.3) | 1 (9.1) |
| | Australia | 3 (6.4) | 3 (8.3) | - |
| Number of authors† | Median | 3 | 3 | 4 |
| | Range | 1–172 | 1–124 | 1–172 |
| Funding | Yes | 32 (68.1) | 23 (63.9) | 9 (81.8) |
| | No | 6 (12.8) | 5 (13.9) | 1 (9.1) |
| | Not reported | 9 (19.1) | 8 (22.2) | 1 (9.1) |
| Funding source‡ | Government | 19 (59.4) | 17 (73.9) | 2 (22.2) |
| | Academic | 15 (46.9) | 9 (39.1) | 6 (66.7) |
| | Non-profit | 14 (43.8) | 9 (39.1) | 5 (55.6) |
| | Unsure | 1 (3.1) | - | 1 (11.1) |
| Ethics approval | Yes | | | |
| | No | 23 (48.9) | 17 (47.2) | 6 (54.5) |
| | Ethics approval not relevant | 10 (21.3) | 8 (22.2) | 2 (18.2) |
| | | 14 (29.8) | 11 (30.6) | 3 (27.3) |

*Data reported at the study level.
†Data reports median and range.
‡Data refers to funded studies only, some studies report multiple funding sources.

48.0%) and experimental studies (52, 29.4%). The remainder of studies captured were data studies for example, re-analysis using previous data (35, 19.8%) or experimental trials (5, 2.8%).

We examined whether the authors of the reproducibility studies included in our synthesis overlapped with the research team of the original studies. To do so, we compared of author lists and examined whether the authors of the reproducibility team self-report their team overlapped or had contact with the original author(s). Sixteen (9.0%) documents had teams that overlapped with the original research team whose study was being replicated, 44 (24.9%) indicated contact with the original team but not authorship overlap, while the remaining 117 (66.1%) studies had no authorship overlap and did not report any contact with the original study authors. Other key findings include that 81 (45.8%) of the studies referred to a registered protocol, although only 41 (23.2%) indicated they used a protocol that was identical to the original study they were reproducing and 28 (23.9%) had both a registered protocol and claimed to be identical to the original study. For 112 (63.3%) of studies the authors indicated that data of the replication studies was publicly available; however, this rate was driven by a few included documents that reported multiple reproduced studies and consistently shared data. Thirty-four of 47 (72.3%) documents included indicated data was not shared. Most studies did not report a primary outcome (134, 75.7%). For studies that did not list a primary outcome, where possible, we extracted the first reported outcome. We thematically grouped the primary/first stated outcomes of the remaining documents into 12 themes, which are presented in Appendix 8. Three of these themes were related to biomedicine or health. We present the data describing the characteristics of the included documents by discipline of the document in *Table 2*.

**Table 2.** Study replication methods characteristics.

| Characteristic | Categories | All discipline studies (N=177) | Biomedicine N (%) | Economics N (%) | Education N (%) | Health sciences* N (%) | Psychology N (%) | Other (mixture of two or more of the above) N (%) |
|---|---|---|---|---|---|---|---|---|
| Did the replication study team specify that they contacted the original study project team? | Yes, the author teams overlapped | 16 (9.0) | 2 (33.3) | - | - | 4 (9.5) | 10 (11.6) | - |
| | Yes, there was contact | 44 (24.9) | - | - | - | 14 (33.3) | 9 (10.5) | 21 (63.6) |
| | No, the teams did not overlap or contact | 117 (66.1) | 4 (66.7) | 5 (100) | 5 (100) | 24 (57.1) | 67 (77.9) | 12 (36.4) |
| Does the replication study refer to a protocol that was registered prior to data collection? | Yes | 81 (45.8) | 2 (33.3) | - | 1 (20.0) | 18 (42.9) | 39 (45.3) | 21 (63.6) |
| | No | 96 (54.2) | 4 (66.7) | 5 (100) | 4 (80.0) | 24 (57.1) | 47 (54.7) | 12 (36.4) |
| Do the authors specify that they used an identical protocol? | Yes | 41 (23.2) | 2 (33.3) | 1 (20.0) | - | 9 (21.4) | 8 (9.3) | 21 (63.6) |
| | No † | 70 (39.5) | 1 (16.7) | 3 (60.0) | 3 (60.0) | 15 (35.7) | 34 (39.5) | 12 (36.64) |
| | Not reported | 64 (36.2) | - | 1 (20.0) | 2 (40.0) | 17 (40.5) | 44 (51.2) | - |
| | Unsure | 2 (1.1) | 2 (33.3) | - | - | - | - | - |
| Does the study indicate that data is shared publicly? | Yes‡ | 112 (63.3) | 2 (33.3) | - | 1 (20.0) | 18 (42.9) | 70 (81.4) | 21 (63.6) |
| | No | 65 (36.7) | 4 (66.7) | 5 (100) | 4 (80.0) | 24 (57.1) | 16 (18.6) | 12 (36.4) |
| What is the study design used? | Data re-analysis | 35 (19.8) | - | 3 (60.0) | 1 (20.0) | 26 (61.9) | 5 (5.8) | |
| | Experimental | 52 (29.4) | 2 (33.3) | 1 (20.0) | - | 10 (23.8) | 39 (45.3) | |
| | Observational | 85 (48.0) | 3 (50.0) | 1 (20.0) | 3 (60.0) | 3 (7.1) | 42 (48.8) | 33 (100) |
| | Trial | 5 (2.8) | 1 (16.7) | - | 1 (20.0) | 3 (7.1) | - | - |
| Did the study specify a primary outcome? | Yes | 43 (24.3) | - | - | - | 26 (61.9) | 13 (15.1) | - |
| | No | 134 (75.7) | 6 (100) | 5 (100) | 5 (100) | 16 (38.1) | 73 (84.9) | 33 (100) |

*One study provided results by outcome not by studies being replicated, in this instance we were unable to determine how the results corresponded to the studies the author listed they replicated so these data are missing.

†In these instances authors specified deviations between their protocol and that of the original research team.

‡This was not verified. We simply recorded what the authors reported. It is possible that self-reported sharing and rates of actual sharing are not identical.

**Table 3.** Reproducibility characteristics of studies replicated overall and across disciplines.

| Characteristic | Categories | Overall | Biomedicine | Economics | Education | Health sciences | Psychology | Other |
|---|---|---|---|---|---|---|---|---|
| How did the authors assess reproducibility? | Effect sizes | 116 (65.5) | 1 (16.7) | 1 (20.0) | 1 (20.0) | 25 (59.5) | 76 (88.4) | 12 (36.4) |
| | Meta analysis of original effect size | 33 (18.6) | 2 (33.3) | - | - | 9 (21.4) | 1 (1.2) | 21 (63.6) |
| | Null hypothesis testing using p-value | 17 (9.6) | 2 (33.3) | 2 (40.0) | - | 5 (11.9) | 8 (9.3) | - |
| | Subjective assessment | 5 (2.8) | - | 1 (20.0) | 2 (40.0) | 2 (4.8) | - | - |
| | Other | 6 (3.4) | 1 (16.7) | 1 (20.0) | 2 (40.0) | 1 (2.4) | 1 (1.2) | - |
| Based on the authors definition of reproducibility, did the study replicate? | Yes | 95 (53.7) | 4 (66.7) | 4 (80.0) | 2 (40.0) | 36 (85.7) | 25 (29.1) | 24 (72.7) |
| | No | 36 (20.3) | 1 (16.7) | 1 (20.0) | 2 (40.0) | 4 (9.5) | 19 (22.1) | 9 (27.3) |
| | Mixed | 8 (4.5) | 1 (16.7) | - | 1 (20.0) | 1 (2.4) | 5 (5.8) | - |
| | Unclear | 38 (21.5) | - | - | - | 1 (2.4) | 37 (43.0) | - |
| Was the p-value reported on the statistical test conducted on the primary outcome? | Yes | 116 (65.5) | 3 (50.0) | 3 (60.0) | 4 (80.0) | 33 (78.6) | 45 (52.3) | 28 (84.8) |
| | No/unclear | 61 (34.5) | 3 (50.0) | 2 (40.0) | 1 (20.0) | 9 (21.4) | 41 (47.7) | 5 (15.2) |

## Definition

*Table 3* provides a summary of definitions for reproducibility. We found that studies related to psychology and health sciences tended to use a comparison of effect sizes to define reproducibility success. The number of included studies across other disciplines was low (<6).

## Prevalence of reproducibility

Of the 177 individual studies reproduced, based on the authors reported definition, 95 (53.7%) reproduced, 36 (20.3%) failed to reproduce, 8 (4.5%) produced mixed results. A further 38 studies (21.5%), 37 of which were from a single included document, could not be assessed due to issues with incomplete or poor-quality reporting. Rates were highest in health sciences (N=36, 85.7%), economics (N=4, 80%), inter-disciplinary studies (N=24, 72.7%). Rates of replication tended to be lower in biomedicine (N=4, 66.7%), education (N=2, 40%), and psychology (N=25, 29.1%). When we removed an included document related to psychology, which presented 37 individual studies but failed to report a reproducibility outcome clearly, rates improved to 51.0% in this discipline. When examining the 35 studies that reported data (re)-analysis projects, rates of reproducibility based on the authors definition were considerably higher (N=31, 88.6%).

Of the 177 individual studies, we were able to extract p-values from 116 (65.5%), of these, reproducibility was statistically significant at the p<0.05 threshold in 82 (70.9%) studies.

## Discussion

The primary objective of this study was to describe the characteristics of reproducibility studies, including how reproducibility is defined and assessed. We found 47 individual documents reporting reproducibility studies in 2018–2019 that met our inclusion criteria. Our included documents reported 177 individual reproducibility studies. Most reproducibility studies were in the disciplines of psychology and health science (>72%), with 86 and 42 studies, respectively. This may suggest unique cultures around reproducibility in distinct disciplines, future research is needed to determine if such differences truly exist given the limitations of our search and approach. Some disciplines have routinely embedded replication as part of the original discovery and validation efforts, for example replications are routinely done for genomics and other -omics findings as part of the original studies rather than as separate efforts. Consistent with previous research (*Makel et al., 2012*; *Sukhtankar, 2017*; *Hubbard and Vetter, 1992*), our findings suggest that overall only a very small fraction of research in any of these discipline published in a given year focuses on reproducing existing research published in previous papers. A recent evaluation of 349 randomly selected research articles from the biomedical literature published in 2015-2018 (*Serghiou et al., 2021*) found that 33 (10%) included a reproducibility component in their research (e.g. validating previously published experiments, running a similar clinical trial in a different population, etc.). However, the vast majority of these efforts would not qualify as separate, independent replication studies in our assessment.

Most of the documents included in our study had corresponding/lead authors who were from the United States (57.4%) and most papers reported receiving funding (68.1%); papers reporting multiple studies (N=9, 81.8%) were more likely to report funding than single replication studies (N=23, 63.9%). Together this may suggest that at least some funders recognize the value of reproducibility studies, and that USA based researchers have taken a leadership role in reproducibility research. We note that just 11 of the 47 papers reported to be reproducing more than one original study, suggesting most reproducibility studies reproduce a single previous study. We note that two of the 'single studies' included reported being part of a larger reproducibility project (e.g. a study part of the broader cancer reproducibility project). Four of 'single studies' reported more than 1 reproduction of the same original article in the document (e.g. different labs reproducing the same experiment). Future qualitative research could shed light on the motivations of researchers to conduct a single versus multiple reproducibility study. This will be important to understand what, if any, supports are needed to facilitate large-scale reproducibility studies.

When we examined the 177 individual studies reproduced in the 47 documents, we found only a minority of them referred to registered protocols. In psychology, rates were highest, with 45.8% of studies referring to a registered protocol. Registered protocols are a core open science practice, they can help to enhance transparency, mitigate publication bias and selective reporting biases (*Nosek*

*et al., 2018*; *Nosek et al., 2019*). Importantly, they may specify what analyses were planned a priori and which were post hoc. Registration would seem especially relevant to reproducibility projects in order to pre-specify approaches to reduce perceptions of bias. We acknowledge, however, that mandates for registration are rare and exist only in particular disciplines and for specific study designs.

There was a wide range of study designs. For example, in psychology observational (e.g. cohort study) and experimental studies dominated, while data analysis studies (e.g. re-analysis) were most prominent in health sciences and economics. Reproducing an observational or experimental study may pose a greater resource challenge as compared to reproducing a data analysis, which may explain the higher rates of reproducibility success observed among data analysis studies. When no new data are generated, it may be difficult in the current research environment, which tends to favor novelty, to publish a re-analysis of existing data that shows the exact same result (*Ebrahim et al., 2014*; *Naudet et al., 2018*).

Across our five disciplines of interest the norm was that author teams did not overlap or contact the team of the original study they were attempting to reproduce. This finding may not be generalizable, because by definition we did not consider documents where the original study was reproduced within the same paper, a practice that is commonplace in many disciplines, for example genomics. Nosek and Errington describe confusion and disagreement that occurred during large scale reproducibility projects they were involved in which produced results that failed to replicate the original findings, calling for original researchers and those conducting reproducibility projects to "argue about what a replication means before you do it". *Nosek and Errington, 2020b* Our finding that teams don't overlap or communicate, suggests that this practice is not typically implemented, despite its potential value to improve reproducibility *Chang and Li, 2015*. Conversely, involvement of the original authors as authors in the reproducibility efforts may increase the impact of allegiance and confirmation biases. In the published experience from the Reproducibility Project: Cancer Biology, a large share of original authors did not respond to efforts to reach them to obtain information about their study (*Errington et al., 2021a*; *Kane and Kimmelman, 2021*; *Errington et al., 2021b*).

Of the 177 individual studies reproduced, based on the authors' reported definition, 53.7% reproduced successfully. When examining definitions for reproducibility, we found that studies related to psychology and health sciences tended to use a comparison of effect sizes to define reproducibility success. The number of included studies across other disciplines was too low to yield meaningful comparison of differences in definitions across disciplines. Rates of reproducibility based on the authors definition were highest in health sciences (N=36/42, 85.7%; 24/33, 72.7%) including 'other' and lower in biomedicine (N=4/6, 66.7%), education (N=2/5, 40%), and psychology (N=25, 29.1%). Low rates in psychology were driven by a single document reporting 37 studies that failed to report outcomes. When this document was removed, rates improved to 51.0% in this discipline. When we applied p-values from the 116 studies where these were reported, 70.7% of studies had p-values less than the commonly used 0.05 threshold. There is an increasing literature of different definitions of reproducibility and 'success' rates will unavoidably depend on how replication is defined (*Errington et al., 2021a*; *Errington et al., 2021b*; *Held et al., 2022*; *Pawel and Held, 2020*).

To our knowledge, this is the first study to provide a broad comparison of the characteristics of explicit reproducibility studies across disciplines. This comparative approach may help to identify features to better support further reproducibility research projects. This study used a formal search strategy, including grey literature searching, to identify potential documents. It is possible that the terminology we used, which was broad to apply across disciplines, may not have captured all potential studies in this area. It is also possible that the databases used do not equally represent the distinct disciplines we investigated, meaning that the searches are not directly comparable cross-disciplinarily. We also were not able to locate the full-text of all included documents which may have impacted on the results. The impact of these missing texts may not have been equal across disciplines. For these reasons, generalizations about disciplines should not be made. A further limitation is that we only considered two years of research. This allowed for a contemporary view on characteristics of replication studies, but it prevented the ability to examine temporal changes. Indeed, several well-known large-scale replication studies would not have been captured in our search. Ultimately, the number of included documents in some disciplines was relatively modest, suggesting that inclusion of articles across a larger timeframe is needed to address the objective to compare more meaningfully across disciplines.

For feasibility we also only extracted information about the primary outcome listed for each paper, or if no primary outcome was specified, the first listed outcome. It is possible that rates of reproducibility differ across outcomes. Future research could consider all outcomes listed. While we conducted our screening and extraction using two reviewers, to foster quality control, the reporting of the studies captured was sometimes extremely poor. This impacted the extraction process as in some cases extraction was challenging and in others resulted in missing data. Further, our screeners and extractors were not naïve to the aims of the study, which may have created implicit bias. Future research could include training coders and extractors who are unaware of the project aims. Collectively these study design decisions and practical challenges present limitations on the overall generalizability of the findings beyond our dataset. Finally, we acknowledge that these explicit 'reproducibility check' documents that we targeted, are only one part of the much larger scientific literature where some reproducibility features may be embedded. Random samples of biomedical papers with empirical data published in 2015–2018 have shown that reproducibility concepts are not uncommon (*Wallach et al., 2018*). In the psychological sciences, similarly 5% of a random sample of 188 papers with empirical data published in 2014–2017 were replications (*Hardwicke et al., 2020*). Conversely, in the social sciences, among a random sample of 156 papers published in 2014–2017, only 2 were replication studies (1%) (*Hardwicke et al., 2020*). Moreover, as mentioned above, some fields explicitly require replications to be included as part of the original publication, and the large and blossoming literature of tens of thousands of meta-analyses (*Ioannidis, 2016*) suggests that for many topics there are multiple studies that address similar enough questions so that meta-analysts would combine them. Eventually, the relative advantages and disadvantages of different replication practices (e.g. reproducibility embedded in the original publication versus done explicitly in a subsequent stage versus done as part of a wider agenda that mixes replication and novel efforts) needs further empirical study in diverse scientific fields.

Our finding that, only about half of the reproducibility studies reproduced across five fields of interest is concerning, though consistent with other studies. These estimates may not necessarily represent appropriately the reproducibility rates of entire fields since the choice of what specific studies to try to replicate may include selection factors that introduce strong bias towards higher or lower replication rates. Moreover, while estimates of reproducibility vary across fields in our modest sample, so too do norms in definitions used to define reproducibility. Choice of these definitions (especially when these definitions are not clear, pre-specified and valid) mayaffect the interpretation of these results to fit various narratives of replication success or failure. This suggests the need for discipline and interdisciplinary specific exchange on how to best approach reproducibility studies. Discussion on definitions for reproducibility, but also about methodological best practices when conducting a reproducibility study (e.g. using registered reports) will help to foster integrity and quality. To ensure reliability, multiple and diverse reproducibility studies with converging evidence are needed. At present, and as illustrated by out sampling, explicit reproducibility studies done as targeted reproducibility checks are rare. To enhance research reliability, reproducibility studies need to be encouraged, incentivized, and supported.

## Additional information

### Funding
No external funding was received for this work.

### Author contributions
Kelly D Cobey, Conceptualization, Data curation, Formal analysis, Supervision, Investigation, Methodology, Writing - original draft, Project administration, Writing – review and editing; Christophe A Fehlmann, Data curation, Formal analysis, Investigation, Visualization, Methodology, Writing – review and editing; Marina Christ Franco, Data curation, Formal analysis, Validation, Investigation, Visualization, Methodology, Writing – review and editing; Ana Patricia Ayala, Lindsey Sikora, Conceptualization, Data curation, Validation, Investigation, Methodology, Project administration, Writing – review and editing; Danielle B Rice, Data curation, Validation, Investigation, Methodology, Writing – review and editing; Chenchen Xu, Manoj M Lalu, Andrew Neitzel, Bea Nguyen, Nino Tsertsvadze, Validation, Investigation, Methodology, Writing – review and editing; John PA Ioannidis, Conceptualization,

Validation, Methodology, Writing – review and editing; Alixe Ménard, Validation, Methodology, Writing – review and editing; David Moher, Conceptualization, Supervision, Validation, Investigation, Methodology, Writing – review and editing

## Author ORCIDs
Kelly D Cobey  http://orcid.org/0000-0003-2797-1686
John PA Ioannidis  http://orcid.org/0000-0003-3118-6859
Manoj M Lalu  http://orcid.org/0000-0002-0322-382X
David Moher  http://orcid.org/0000-0003-2434-4206

## Decision letter and Author response
Decision letter https://doi.org/10.7554/eLife.78518.sa1
Author response https://doi.org/10.7554/eLife.78518.sa2

---

# Additional files

## Supplementary files
• MDAR checklist

## Data availability
Data and materials are available on the Open Science Framework (https://osf.io/wn7gm/).

The following dataset was generated:

| Author(s) | Year | Dataset title | Dataset URL | Database and Identifier |
|---|---|---|---|---|
| Cobey KD | 2022 | Scoping review data on reproducibility of research in a 2018-2019 multi-discipline sample | https://osf.io/x6vs3/ | Open Science Framework, 10.17605/OSF.IO/WN7GM |

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

## Appendix 1

## Operationalized definitions of included disciplines

| Discipline | Definition |
|---|---|
| Health sciences | Nutritional sciences, physiotherapy, kinesiology, rehabilitation, speech language pathology, physiology, nursing, midwifery, occupational therapy, social work, medicine and all its specialties, public health, population health, global health, pathology, laboratory medicine, optometry, health services research, |
| Biomedicine | Neuroscience, pharmacology, radiation therapy, dentistry, health management, epidemiology, virology, biomedicine, clinical engineering, biomedical engineering, genetics, |
| Education | Higher education, adult education, K-12 education, medical education, health professions education |
| Psychology | All specializations in Psychology, including but not limited to: clinical psychology, group psychology, psychotherapy, counselling, industrial psychology, cognitive psychology, forensic psychology, health psychology, neuropsychology, occupational psychology, social psychology |
| Economics | Microeconomics, macroeconomics, behavioural economics, econometrics, international economics, economic development, agricultural economics, ecological economics, environmental economics, natural resource economics, economic geography, location economics, real estate economics, regional economics, rural economics, transportation economics, urban economics, capitalist systems, comparative economic systems, developmental state, economic systems, transitional economies, economic history, industrial organization |

# Appendix 2

## Search Strategy

1. exp "reproducibility of results"/
2. ((reproduc* or replicat* or reliabilit* or repeat* or repetition) adj2 (result* or research* or test*)). tw,kf.
3. (face adj validit*).tw,kf.
4. (test adj reliabilit*).tw,kf.
5. or/1–4
6. prevalence/
7. (prevalen* or rate or rates or recur* or reoccuren*).tw,kf.
8. 6 or 7
9. 5 and 8
10. limit 9 to (yr="2018–2019" and english)

## PRISMA-S Checklist

| Section/topic | # | Checklist item | Location(s) Reported |
|---|---|---|---|
| INFORMATION SOURCES AND METHODS | | | |
| Database name | 1 | Name each individual database searched, stating the platform for each. | ~line 185–190 |
| Multi-database searching | 2 | If databases were searched simultaneously on a single platform, state the name of the platform, listing all of the databases searched. | n/a |
| Study registries | 3 | List any study registries searched. | n/a |
| Online resources and browsing | 4 | Describe any online or print source purposefully searched or browsed (e.g., tables of contents, print conference proceedings, web sites), and how this was done. | ~line 195 |
| Citation searching | 5 | Indicate whether cited references or citing references were examined, and describe any methods used for locating cited/citing references (e.g., browsing reference lists, using a citation index, setting up email alerts for references citing included studies). | ~line 190–194 |
| Contacts | 6 | Indicate whether additional studies or data were sought by contacting authors, experts, manufacturers, or others. | See appendix 3 |
| Other methods | 7 | Describe any additional information sources or search methods used. | n/a |
| SEARCH STRATEGIES | | | |
| Full search strategies | 8 | Include the search strategies for each database and information source, copied and pasted exactly as run. | See appendix 2 |
| Limits and restrictions | 9 | Specify that no limits were used, or describe any limits or restrictions applied to a search (e.g., date or time period, language, study design) and provide justification for their use. | |
| Search filters | 10 | Indicate whether published search filters were used (as originally designed or modified), and if so, cite the filter(s) used. | n/a |
| Prior work | 11 | Indicate when search strategies from other literature reviews were adapted or reused for a substantive part or all of the search, citing the previous review(s). | n/a |
| Updates | 12 | Report the methods used to update the search(es) (e.g., rerunning searches, email alerts). | n/a |
| Dates of searches | 13 | For each search strategy, provide the date when the last search occurred. | ~line 190–194 |
| PEER REVIEW | | | |
| Peer review | 14 | Describe any search peer review process. | ~line 182 |

*Continued on next page*

*Continued*

| Section/topic | # | Checklist item | Location(s) Reported |
|---|---|---|---|
| MANAGING RECORDS | | | |
| Total Records | 15 | Document the total number of records identified from each database and other information sources. | Figure 1 |
| Deduplication | 16 | Describe the processes and any software used to deduplicate records from multiple database searches and other information sources. | ~line 200–203 |

PRISMA-S: An Extension to the PRISMA Statement for Reporting Literature Searches in Systematic Reviews. Rethlefsen ML, Kirtley S, Waffenschmidt S, Ayala AP, Moher D, Page MJ, Koffel JB, PRISMA-S Group. Last updated February 27, 2020.

## Appendix 3

### Grey literature search approach

1. Search reference lists of articles included for data extraction.
2. Forward/backward citation analysis of articles included for data extraction in Scopus and Web of Science (platforms including Science Citation Index Expanded (SCI-EXPANDED) -–1900-present and Social Sciences Citation Index (SSCI) -–1900-present).
3. Google Scholar search as follows: "Reproducibility" limited to the years 2018–2019.

Search of the following preprint servers: OpenScience Framework (OSF) including OSF Preprints, bioRxiv, EdArXiv, MediArXiv, NutriXiv, PeerJ, Preprints.org, PsyArXiv and SocArXiv, NBER Working Papers, Munich Personal RePEc Archive

## Appendix 4

## Level 1 Screening form item and criteria

| 1. ISSUE: Does the study self-report to be a replication of previous quantitative research? (yes/no/unsure) | |
| --- | --- |
| **INCLUDE** | **EXCLUDE** |
| All research articles containing one or more replications of quantitative research studies. | All research articles not describing a replication study All research articles describing exclusively qualitative replication studies |

## Appendix 5

## Level 2 Screening form

1. LANGUAGE – Is this study in English? (yes/no/unsure)

| INCLUDE | EXCLUDE |
| --- | --- |
| Studies written in English. | Studies written in any other language that is not English. |

2. DATE– Is this study published in 2018 or 2019? Use the most recent year stated on the publication (yes/no/unsure)

| INCLUDE | EXCLUDE |
| --- | --- |
| Studies published in 2018 or 2019. | Studies publisher in any other year. EPub ahead of print. |

3. PUBLICATION TYPE – Is this the right publication type? (yes/no/unsure)

| INCLUDE | EXCLUDE |
| --- | --- |
| Original research articles describing quantitative research. | <ul><li>Narrative reviews</li><li>Scoping reviews</li><li>Systematic reviews</li><li>Realist reviews</li><li>Mapping reviews</li><li>Literature reviews</li><li>Rapid reviews</li><li>Meta-Analysis</li><li>Overview or reviews</li><li>Umbrella reviews</li><li>In short any type of literature review or synthesis should be excluded.</li><li>Conference proceedings</li><li>Book chapters</li><li>Editorials, letters to the editor, commentaries</li><li>Opinion pieces</li><li>Case reports</li><li>Case control studies</li><li>Case series</li><li>Protocols</li><li>Guidelines</li><li>Web pages</li><li>Thesis projects</li><li>Policy documents</li><li>All exclusively qualitative research studies</li></ul> |

4. ISSUE: Does the study self-report to be a replication of previous quantitative research? (yes/no/unsure)

| INCLUDE | EXCLUDE |
| --- | --- |
| All research articles containing one or more replications of quantitative research studies. | All research articles not describing a replication study. All research articles describing exclusively qualitative replication studies |

5. DISCIPLINE – Does the study focus education, economics, psychology, biomedicine or health sciences? (yes/no/unsure)

| INCLUDE | EXCLUDE |
| --- | --- |
| All research that is related to the disciplines of education, economics, psychology, biomedicine or health sciences. (*Table 1* info to be provided). | All research that is not related to education, economics, psychology, biomedicine or health sciences. Exclude research related to complementary and alternative medicine. |

## Appendix 6

### Level 3 extraction form

1. Year of publication (use the most recent year stated on the document):
2. Name of the journal/outlet the document is published in (Do not use abbreviations):
3. Corresponding author e-mail (If there is more than one corresponding author indicated, extract the first listed author only. Extract the name in the format: Initial Surname, e.g., D Moher. If there is more than one e-mail listed, extract the first listed e-mail only.):
4. Country of corresponding author affiliation (If there is more than one corresponding author indicated, extract the first listed corresponding author only. If there is more than one affiliation listed for this individual, extract from the first affiliation only):
5. How many authors are named on the document?
6. Does the study report a funding source (yes, no)
   a. If yes, which type of funder. Check all that apply. (Government, Academic, Industry, Non-Profit, Other, Can't tell)
7. Did the study report ethics approval (yes, no, ethics approval not relevant)
8. Did the study recruit human participants or use animal participants? (yes, no)
   a. If yes; Does the study involve humans or animals? (humans, animals)
   b. If human; Please specify how many were involved? (use the total number of participants enrolled, not necessarily the number analyzed):
   c. If animal; Please specify how many were involved? (use the total number of animals included, not necessarily the number analyzed):
9. Did the replication study team specify that they contacted the original study project team? (Yes, the author teams were reported to overlap; Yes, there was contact but author teams do not overlap; No, the replication team did not report any interaction)
10. Does the replication study refer to a protocol that was registered prior to data collection? (yes, no)
11. Does the study indicate that data is shared publicly? (yes, no)
12. How many quantitative replication studies are reported in the paper? (One study, more than one study).

Note: If more than one study is reported, we will extract information from each quantitative study using the following questions.

13. What is the study design used: (observational study; clinical trial; experimental; data analysis, other):
14. Did the study specify a primary outcome being? (Yes/No).
15. If a primary outcome was stated, what was it? If a primary outcome was not stated, please extract that first stated outcome described in the study results section (note these will be thematically grouped).
16. What discipline does this work best fit in? (Economics, Education, Psychology, Health Sciences, Biomedicine, Other)
17. How did the authors of the study assess reproducibility?

**Evaluating against the null hypothesis:** determining whether the replication showered a statistically significant effect, in the same direction as the original study, with a *P*-value <0.05.
**Effect sizes:** Evaluating replication effect against original effect size to examine for differences.
**Meta-analysis of original effect size:** Evaluates effect sizes considering variance and of 95% confidence intervals.
**Subjective assessment of replication:** An evaluation made by the research team as to whether they were successful in replicating the study findings.
**Other, please specify**.
**Unclear**

18. Based on the authors definition of reproducibility, did the study replicate? (yes, no, mixed)

19. What was the p-value reported on the statistical test conducted on the main outcome? (value:; not reported)

20. What was the effect size reported on the statistical test conducted on the main outcome? Size: Measure:

# Appendix 7

## List of included documents

| ID | Year | Journal | Corresponding Author | Funding | Number of studies replicated | Discipline |
|---|---|---|---|---|---|---|
| 20131 | 2018 | Advances in Methods and Practices in Psychological Science | RJ McCarthy | Yes | One study | psychology |
| 20130 | 2018 | Advances in Methods and Practices in Psychological Science | B Verschuere | Yes | One study | psychology |
| 20128 | 2018 | Association for Psychological Science | M O'Donnell | Yes | One study | psychology |
| 20126 | 2019 | Association for Psychological Science | CJ Soto | Yes | More than one study | psychology |
| 20125 | 2018 | Psychological Science | TW Watts | Yes | One study | psychology |
| 20124 | 2018 | eLife | MS Nieuwland | Yes | One study | psychology |
| 20122 | 2019 | Journal of Environmental Psychology | S Van der Linden | Not reported | One study | psychology |
| 20120 | 2019 | The European Political Science Association | A Coppock | Yes | More than one study | other |
| 20119 | 2018 | Finance Research Letters | dirk.baur@uwa.edu.au | Not reported | One study | economics |
| 20094 | 2019 | Journal of Economic Psychology | AK Shah | Yes | More than one study | psychology |
| 20048 | 2018 | bioRxiv | X Zhang | Yes | One study | biomedicine |
| 20001 | 2018 | Royal society open science | T Schuwerk | Yes | More than one study | psychology |
| 10805 | 2018 | Rehabilitation Counseling Bulletin | BN Philips | Yes | One study | economics |
| 20000 | 2018 | Advances in Methods and Practices in Psychological Science | RA Klein | Yes | More than one study | psychology |
| 13598 | 2018 | Molecular Neurobiology | A Chan | Yes | One study | biomedicine |
| 13165 | 2018 | Journal of Contemporary Criminal Justice | JP Stamatel | No | One study | psychology |
| 12923 | 2019 | PLOS One | LM Smith | Yes | One study | health sciences |
| 12287 | 2019 | J Autism Dev Disord | L K Fung | Yes | One study | psychology |
| 11535 | 2018 | Journal of Obsessive-Compulsive and Related Disorders | EN Riise | No | One study | psychology |
| 11456 | 2018 | eLife | J Repass | Yes | One study | biomedicine |
| 11003 | 2018 | Journal of Health Economics | D Powell | Yes | One study | health sciences |
| 10567 | 2019 | Personality and Individual Differences | JJ McGinley | No | One study | health sciences |
| 10555 | 2019 | J Nerv Ment Dis | G Parker | Yes | One study | psychology |
| 9886 | 2018 | The BMJ | J P A Ioannidis | No | More than one study | health sciences |
| 9505 | 2019 | BMC Geriatrics | S E Straus | Yes | One study | health sciences |
| 9193 | 2018 | Journal for Research in Mathematics Education | K Melhuish | Not reported | More than one study | education |
| 9011 | 2018 | Journal of Contemporary Criminal Justice | CD Maxwell | Yes | One study | psychology |
| 8574 | 2018 | Public Opinion Quarterly | J A Krosnick | Yes | One study | economics |
| 6213 | 2019 | Psychophysiology | M Arns | No | One study | biomedicine |
| 5825 | 2019 | BMC Geriatrics | J.Holroyd-Leduc | Yes | One study | health sciences |
| 5539 | 2018 | Empirical Economics | B.Hayo | Not reported | One study | economics |
| 5458 | 2018 | Reproducing Public Health Services and Systems Research | J K Harris | Yes | More than one study | health sciences |
| 5138 | 2019 | Behavior Therapy | E J Wolf | Yes | One study | psychology |

*Continued on next page*

*Continued*

| ID | Year | Journal | Corresponding Author | Funding | Number of studies replicated | Discipline |
|---|---|---|---|---|---|---|
| 5133 | 2019 | Brain, Behavior, and Immunity | FR Guerini | Yes | One study | biomedicine |
| 5100 | 2019 | European Neuropsychopharmacology | R Lanzenberger | Yes | One study | biomedicine |
| 4794 | 2018 | American Psychological Association | R J Giuliano | Not reported | One study | health sciences |
| 3962 | 2019 | Journal of Applied Behaviour Analysis | G Rooker | Not reported | One study | health sciences |
| 3677 | 2019 | Journal of Pediatric Psychology | B D Earp | Yes | One study | psychology |
| 3574 | 2019 | Personnel Psychology | G F Dreher | Not reported | One study | psychology |
| 3138 | 2018 | Journal of Second Language Writing | C de Kleine | Yes | One study | education |
| 2310 | 2019 | PLOS ONE | B Chen | Yes | More than one study | health sciences |
| 1959 | 2018 | Nature Human Behaviour | BA Nosek | Yes | More than one study | other |
| 1837 | 2018 | Oxford Bulletin Of Economics and Statistics | D Buncic | Not reported | One study | economics |
| 1681 | 2018 | Cortex | SG Brederoo | Yes | More than one study | health sciences |
| 1477 | 2019 | Frontiers in Psychology | M Boch | Yes | One study | psychology |
| 727 | 2018 | Archives of Clinical Neuropsychology | P Armistead-Jehle | No | One study | health sciences |
| 584 | 2018 | Australian Psychologist | RJ Brunton | Not reported | One study | health sciences |

## Appendix 8

## Thematic groups of primary outcomes of studies replicated

| No. | Theme | N (%) |
|---|---|---|
| 1 | Clinical and biological outcomes | 19 (10.7) |
| 2 | Public health | 5 (2.8) |
| 3 | Mental health and wellbeing | 6 (3.4) |
| 4 | Criminology | 5 (2.8) |
| 5 | Economics | 8 (4.5) |
| 6 | Individual differences | 58 (32.8) |
| 7 | Visual cognition | 11 (6.2) |
| 8 | Morality | 5 (2.8) |
| 9 | Score and performance | 11 (6.2) |
| 10 | Political views | 11 (6.2) |
| 11 | Not reported/unclear | 30 (17.0) |
| 12 | Other | 8 (4.5) |

