## [Editor Report]

It has been recognized since the beginning of science that science can always be made more rigorous. Indeed, it is part of the ethos and very nature of the scientific method and the scientific attitude, as Lee McIntyre describes in his brilliant book by that title, to be constantly striving for improvements in rigor. Yet, we know that there are breaches in rigor, reproducibility, and transparency of research conduct and reporting. Such breaches have been highlighted more intensively, or at least so it seems, for more than a decade. The field recognizes that we need to go beyond platitudinous recognition that there is always opportunity for improvement in rigor and that such improvements are vital, to identifying those key leverage points where efforts can have the most positive near-term effects. Identifying domains in which reproducibility is greater or lesser than in other domains can aid in that regard. Thus, this article represents a constructive step in identifying key opportunities for bettering our science and that is something that every scientist can stand behind.

---

## [Decision Letter]

**Decision letter after peer review:**

[Editors’ note: the authors submitted for reconsideration following the decision after peer review. What follows is the decision letter after the first round of review.]

Thank you for submitting your article "Epidemiological characteristics and prevalence rates of research reproducibility across disciplines: A scoping review" for consideration by *eLife*. Your article has been reviewed by 3 peer reviewers, and the evaluation has been overseen by a Reviewing Editor and Mone Zaidi as the Senior Editor. The following individuals involved in review of your submission have agreed to reveal their identity: Colby J Vorland (Reviewer #1); Arthur Lupia (Reviewer #2); Jon Agley (Reviewer #3).

I believe that the reviewers' comments are clear and should be helpful to you.

*Reviewer #1 (Recommendations for the authors):*

Cobey et al. conducted an exhaustive survey of a number of five disciplines to catalogue and characterize replications. The authors employed best practices in performing their review and reporting, and the results provide an important snapshot of replications. Kudos to the authors for their transparent practices – preregistering their protocol, and specifying protocol amendments.

A limitation of this work is that it only surveys 2018-2019, and many replication projects were not published then, which limits comparisons across disciplines. This is understandable given the herculean screening task for just these two years, and the authors do emphasize this limitation in their discussion. Including the years 2018-2019 in the title seems appropriate to make this clear in searches.

The manuscript is generally well written, but throughout the text the authors use the terms 'reproducibility' and 'replicability' interchangeably (sometimes within the same sentence), which is confusing. Even more so is that the authors attempt to distinguish the definitions of each in the introduction:

Line 102: [Defining 'reproducibility'] "Here, we loosely use Nosek and Errington's definition: "a study for which any outcome would be considered diagnostic evidence about a claim from prior research"".

– However, this quote refers to the term 'replication'. Here is the full quote: "Replication is a study for which any outcome would be considered diagnostic evidence about a claim from prior research." An additional quote from that paper defines reproducibility and replication like so: "Credibility of scientific claims is established with evidence for their replicability using new data [1]. This is distinct from retesting a claim using the same analyses and same data (usually referred to as reproducibility or computational reproducibility) …" These definitions are in alignment with the 2019 NASEM report (https://nap.nationalacademies.org/catalog/25303/reproducibility-and-replicability-in-science). As far as I can understand, the authors studied *both* concepts in this paper; most published reports were replications, but it is noted on line 330 that "The remainder of studies captured were data studies e.g., re-analysis using previous data (35, 19.8%)…". If it is the case and that this means that these data studies re-analyzed the *same* dataset as the original work, the manuscript should be rewritten to distinguish replication from reproducibility throughout. The authors may wish to note the fact that different disciplines may use these terms in different ways (e.g., expand the discussion of reference #1), and that reproducibility is sometimes used to refer to transparency, and in other contexts replicability.

In the introduction, the authors note the percentages of studies that replicate for various replication projects. Yet, that is taking each project's definition of what a successful replication is. As the authors point out on line 131, some may define in different ways and thus simplified conclusions are difficult. This point could be moved earlier to emphasize that this is the case for all of these projects.

The data statement states: "Data and materials are available on the Open Science Framework (https://osf.io/wn7gm/)." I see the protocol there, but no extracted data; will these be added?

Could some of the results in tables 1-3 also be presented graphically?

*Reviewer #2 (Recommendations for the authors):*

This article offers an important discussion, and datapoint, relevant to helping scientific researchers more closely reconcile what they observe with what they claim. There is tremendous potential public and scientific value in the endeavor. The value arises from the fact that many academic reward systems are tilted towards mass production of statistically significant claims and away from truer representations of what these claims mean to readers or end users. While there is greater recognition of these problems, and many attempts to improve practices, this article raises tough questions about how well actual practice aligns with publicly stated RR goals. The authors seek a constructive way forward. They offer an empirical analysis that documents and compares practices across several disciplines. Tables 2 and 3 are particularly instructive and a model for future work of this kind.

My biggest concern with the paper is that there is a gap between the article's findings and several generalizations that are made. I find the method and empirical results interesting, but have limited confidence in how much these claims generalize – even to representative samples of articles in the fields on which the paper focuses. Edits that more closely align the text with the observations will help readers more accurately interpret the authors' findings.

For example, the authors list a series of discipline-specific article databases that they use to identify articles for comparison. If a goal is to make discipline-wide generalizations from these set of articles, it would help readers to know more about the comparability of the databases. For example, are the databases equally representative of the fields that they are characterized as covering (i.e., do some of the databases favor subfields within a discipline or lack full coverage in other ways)? As the article is written, the reader is asked to take this premise on faith. The truth-value of this premise is not apparent to me. If there is no empirical or structural basis for assuming direct comparability, the fact should be noted, cross-disciplinary comparisons or conclusions should reference the caveat, and generalizations beyond this caveat should not be made.

Another aspect of the analysis that may impede generalization is the decision to list the first reported RR outcome in studies that do not list a primary outcome. Given known incentives that lead to publication biases, and against null results, isn't it likely that first reported outcomes are more likely to be non-replications? If this is a possibility, could they compare RR success rates from first reported outcomes to last-reported outcomes? If there is no difference, the fact can be noted and a particular type of generalization becomes possible. If there is a difference, then the caveat should be added in subsequent claims.

On a related note, the authors conclude that RR frequencies and success rates are modest. But what is the relevant base rate? This standard may be easier to define for RR outcomes. On the optimal number of RR attempts, I think that there is less of a consensus. To state the question in a leading manner, if one replication/reproduction is better than none, why aren't two better than one? This question, and others like it, imply that observed rates of RR attempts that are less than 100% may not be suboptimal in a broader sense.

On a different note, I am concerned about how the coding was done. Specifically, it appears that co-authors served as "screeners" of how to categorize certain articles and article attributes. In other words, they made decision that influenced the data that they were analyzing. The risk is that people who know the hypotheses and desired outcomes implicitly bring that knowledge to coding decisions. An established best practice in fields where this type of coding is done is to first train independent coders who are unaware of the researchers; hypotheses and then conduct rigorous inter-coder reliability assessments to document the extent to which the combination of coders and categorical framework produce coding outcomes that parallel the underlying conceptual framework. Such practices increase the likelihood that data generating processes are independent of hypothesis evaluation processes.

For these reasons, I am very interested in the questions that the authors ask, and what they find, but I am not convinced that a number of the empirical claims pertaining to comparisons and magnitudes will generalize even to larger populations of articles in the stated disciplines.

I would like to see the Discussion section rewritten to focus on what the findings, and methodological challenges, mean for future work. In the current version, there are a number of speculative claims and generalizations that do not follow from the empirical work in this paper. These facts and some variation in the focus of the text in that section make the discussion longer than it needs to be and may have the effect of diminishing the excellent work that was done in the earlier pages.

*Reviewer #3 (Recommendations for the authors):*

The authors provide a fairly clear and accurate summary of the current questions around reproducibility in the scientific literature. I was impressed that they thoughtfully included a description of the areas where their utilized methodology was different than what they had proposed in their protocol and explained the reasons for those differences. That kind of documentation is valuable and fosters transparency in the research process.

The results of their analysis are provided in both narrative format and in tables and appendices. The authors effectively characterize the different, and sometimes difficult, decisions made in parsing the results of their search. While much attention has been paid to replication and reproducibility in recent years, the nature of the results reflects the reality that the replication studies themselves may have reporting issues, and it may not always be possible to ascertain how to link the results of such studies with the original work. The study results meaningfully add to the current, small body of literature examining the meta-scientific issues of reproducibility and replication of scientific findings.

The paper is cautious in its approach to interpretation, appropriately using language such as "this may suggest that…" to better ensure that readers do not draw inappropriately firm conclusions. This is a descriptive paper, and so readers, like the authors, would best be served by limiting strong inferences based on the findings. At the same time, the authors helpfully suggest numerous meaningful pathways forward to advance reproducibility research in multiple scientific domains. Such studies would in turn facilitate more powerful inferences.

In preparing this review, the substantive majority of recommendations that I had for the authors related to points of clarity conveyed in private comments rather than overt weaknesses or concerns.

LL50-52: The meaning of this sentence is not entirely clear (though the paragraph can still be understood when skipping the sentence). Please consider rewriting to capture your point more closely.

LL106-108: Since that analysis was published in 2014, and given the recent (albeit anecdotal) increased emphasis on reproducibility, it may be worth noting the date of the study in the text itself (or the termination date of the search used for the 0.13% statistic).

L126: In this context, "some effect in inbreeding" is unclear.

LL135-136: There may be value in disentangling the link between reproducibility and harms in the Introduction more explicitly. In the cited study (Le Noury et al), medications were utilized in a trial where reanalysis indicated that they were unlikely to produce a clinical benefit and each resulted in plausibly increased risk of harm. But it was not the failure to replicate, per se, that led to possible harms. Rather, the concern was clinical decision making predicated on the results of the original study, the results of which "stood" pending reanalysis. This may seem like a pedantic point but I think it speaks to the role not only of the original study itself, but also the degree to which decisions were made based on the findings of a single study.

L240: Can you provide a short clarification of what you mean by calibration test – such as inserting a parenthetical example "…performed a calibration test (e.g., did X) prior to…"

Table 1: The box indicating a cross section of number of authors and all studies appears to have inaccurate/incomplete information.

Table 1: The way the data are structured is a little unclear. For example, for "All Studies" the "discipline" row uses a denominator of total replication studies, whereas "year of publication" in the same column uses a denominator of the number of papers. While I see why this was done, I think additional clarity or uniformity in how the data are presented would be helpful.

LL300-301: Could you briefly indicate whether you think that excluding documents unavailable in full via your library may have affected the results (e.g., was this 11 quantitative replications, as assessed by abstracts? or was it a mixed bag?).

L384: Does the count for psychology include the three dozen or so replication studies that were unusable for this paper? If so, do the authors think that their inclusion here serves to accurately represent the replication efforts in psychology or may artificially inflate the frequency? I don't have the perspective to know which is the case, but I think the question is worth considering.

LL397-398: Would you be willing to share any information (if it exists) about whether independent and separate replications differ from those embedded within experimental protocols in meaningful ways (not just in terms of the obvious process differences)?

L417: Are you referring to registered protocols for the replication studies? Or registered protocols for the original study that can be used in completing the replication?

LL430-432: It might be helpful for readers to be briefly informed about what "the current environment" means in practice, as there is a lot of possible variability. For example, I might assume that there is an overemphasis on novelty at the desk editing stage, but it's not clear whether the authors were even thinking of that in particular.

LL434-448: There is a lot of interesting discussion in this paragraph that gets somewhat "jumbled together" in this paragraph. First, I'm not sure that it makes sense to characterize the finding as biased in L436. Rather, you found that for replication studies that were not part of the same paper, lack of overlap was common, and you do not make any assertions about studies where that was not the case (but presume the overlap might be higher – and in fact would by definition be nearly 100% since authors do not change partway through papers). Second, the larger question about what constitutes a replication is related to, but seemingly separate from, the initial discussion. Even if the authors agree on what a replication would look like for a specific study, it may not be coherent in the context of the broader field's common understanding of reproducibility.

[Editors’ note: further revisions were suggested prior to acceptance, as described below.]

Thank you for resubmitting your work entitled "Epidemiological characteristics and prevalence rates of research reproducibility across disciplines: A scoping review of articles published in 2018-2019" for further consideration by *eLife*. Your revised article has been evaluated by Mone Zaidi (Senior Editor) and a Reviewing Editor.

The manuscript has been improved but there are some modest remaining issues that need to be addressed, as outlined below:

*Reviewer #3 (Recommendations for the authors):*

I would like to thank the authors for their response to my original comments. In most cases, I found that the authors' revisions were responsive to my concerns. In the instances where I did not find that to be entirely the case, I have noted it below using the new line numbers.

L126: Thank you for revising the line about inbreeding. However, I think that the replacement phrase is still a little unclear. The sentence is structured as an 'either/or' and the first part suggests familiarity with the study method may increase likelihood of reproducing the same findings. So I am not sure how "researcher familiarity" functions as the alternative. The sense I get is that the authors may be hedging around suggesting that in some cases, the author's approach to research (perhaps unusually rigorous, or unusually sloppy) may carry over between studies and explain some of the variability in findings. However, that is an assumption. In any case, the authors might benefit from being very direct about their theory here.

L110: The phrase "…and some disciplines do worse in this regard" might be interpreted to have two meanings. I interpret it to mean that some disciplines have fewer studies where reproductions have been attempted. However, "do worse" might also be taken to mean that some disciplines' studies are less likely to successfully have their results reproduced, which is a different concept entirely.

LL404-405: I appreciate this revision made in response to reviewer #1. I am unsure about whether an assertion of "unique cultures" is significantly different than the original assertion of which cultures have it as a normative value. I think the line may be fine to retain, but should be constrained with a caveat e.g., "This may suggest unique cultures around reproducibility across different disciplines, but further study is needed to determine whether this is truly the case, and our study should not be taken as proof of differences between disciplines."

---

## [Author Response]

[Editors’ note: The authors appealed the original decision. What follows is the authors’ response to the first round of review.]

Reviewer #1 (Recommendations for the authors):Cobey et al. conducted an exhaustive survey of a number of five disciplines to catalogue and characterize replications. The authors employed best practices in performing their review and reporting, and the results provide an important snapshot of replications. Kudos to the authors for their transparent practices – preregistering their protocol, and specifying protocol amendments.A major limitation of this work is that it only surveys 2018-2019, and many replication projects were not published then, which limits comparisons across disciplines. This is understandable given the herculean screening task for just these two years, and the authors do emphasize this limitation in their discussion. Including the years 2018-2019 in the title seems appropriate to make this clear in searches.

We have updated the title to incorporate this feedback. It now reads: Epidemiological characteristics and prevalence rates of research reproducibility across disciplines: A scoping review of articles published in 2018-2019.

The manuscript is generally well written, but throughout the text the authors use the terms 'reproducibility' and 'replicability' interchangeably (sometimes within the same sentence), which is confusing. Even more so is that the authors attempt to distinguish the definitions of each in the introduction:Line 102: [Defining 'reproducibility'] "Here, we loosely use Nosek and Errington's definition: "a study for which any outcome would be considered diagnostic evidence about a claim from prior research"".– However, this quote refers to the term 'replication'. Here is the full quote: "Replication is a study for which any outcome would be considered diagnostic evidence about a claim from prior research." An additional quote from that paper defines reproducibility and replication like so: "Credibility of scientific claims is established with evidence for their replicability using new data [1]. This is distinct from retesting a claim using the same analyses and same data (usually referred to as reproducibility or computational reproducibility) …" These definitions are in alignment with the 2019 NASEM report (https://nap.nationalacademies.org/catalog/25303/reproducibility-and-replicability-in-science). As far as I can understand, the authors studied *both* concepts in this paper; most published reports were replications, but it is noted on line 330 that "The remainder of studies captured were data studies e.g., re-analysis using previous data (35, 19.8%)…". If it is the case and that this means that these data studies re-analyzed the *same* dataset as the original work, the manuscript should be rewritten to distinguish replication from reproducibility throughout. The authors may wish to note the fact that different disciplines may use these terms in different ways (e.g., expand the discussion of reference #1), and that reproducibility is sometimes used to refer to transparency, and in other contexts replicability.

Thanks for this thoughtful reflection. We agree the terminology could be clarified further. We have adjusted this section, it now reads:

“Reproducibility is a central tenant of research. Reproducing previously published studies to determine if results are consistent helps us to discern discoveries from false leads. The lexicon around the terms reproducibility and replication is diverse and poorly defined and may differ between disciplines ^1^*.* Here, we loosely use Nosek and Errington’s definition for a successful replication, namely: “a study for which any outcome would be considered diagnostic evidence about a claim from prior research”^2^*.”*

In the introduction, the authors note the percentages of studies that replicate for various replication projects. Yet, that is taking each project's definition of what a successful replication is. As the authors point out on line 131, some may define in different ways and thus simplified conclusions are difficult. This point could be moved earlier to emphasize that this is the case for all of these projects.

We have taken on this suggestion by adding the line “Comparison of rates of replication prove challenging because results will depend on the definition of replication success.” further ahead in the paper, around line 118.

The data statement states: "Data and materials are available on the Open Science Framework (https://osf.io/wn7gm/)." I see the protocol there, but no extracted data; will these be added?Could some of the results in tables 1-3 also be presented graphically?

All study data and materials have been made available at this link now.

Reviewer #2 (Recommendations for the authors):This article offers an important discussion, and datapoint, relevant to helping scientific researchers more closely reconcile what they observe with what they claim. There is tremendous potential public and scientific value in the endeavor. The value arises from the fact that many academic reward systems are tilted towards mass production of statistically significant claims and away from truer representations of what these claims mean to readers or end users. While there is greater recognition of these problems, and many attempts to improve practices, this article raises tough questions about how well actual practice aligns with publicly stated RR goals. The authors seek a constructive way forward. They offer an empirical analysis that documents and compares practices across several disciplines. Tables 2 and 3 are particularly instructive and a model for future work of this kind.My biggest concern with the paper is that there is a gap between the article's findings and several generalizations that are made. I find the method and empirical results interesting, but have limited confidence in how much these claims generalize – even to representative samples of articles in the fields on which the paper focuses. Edits that more closely align the text with the observations will help readers more accurately interpret the authors' findings.For example, the authors list a series of discipline-specific article databases that they use to identify articles for comparison. If a goal is to make discipline-wide generalizations from these set of articles, it would help readers to know more about the comparability of the databases. For example, are the databases equally representative of the fields that they are characterized as covering (i.e., do some of the databases favor subfields within a discipline or lack full coverage in other ways)? As the article is written, the reader is asked to take this premise on faith. The truth-value of this premise is not apparent to me. If there is no empirical or structural basis for assuming direct comparability, the fact should be noted, cross-disciplinary comparisons or conclusions should reference the caveat, and generalizations beyond this caveat should not be made.

Thanks for this comment. We have modified the paper to address this in the discussion. We have added: “It is also possible that the databases used do not equally represent the distinct disciplines we investigated, meaning that the searches are not directly comparable cross-disciplinarily”.

Another aspect of the analysis that may impede generalization is the decision to list the first reported RR outcome in studies that do not list a primary outcome. Given known incentives that lead to publication biases, and against null results, isn't it likely that first reported outcomes are more likely to be non-replications? If this is a possibility, could they compare RR success rates from first reported outcomes to last-reported outcomes? If there is no difference, the fact can be noted and a particular type of generalization becomes possible. If there is a difference, then the caveat should be added in subsequent claims.

We have now noted this possibility in the discussion. We have added: “For feasibility we also only extracted information about the primary outcome listed for each paper, or if no primary outcome was specified, the first listed outcome. It is possible that rates of replication differ across outcomes. Future research could consider all outcomes listed.”

On a related note, the authors conclude that RR frequencies and success rates are modest. But what is the relevant base rate? This standard may be easier to define for RR outcomes. On the optimal number of RR attempts, I think that there is less of a consensus. To state the question in a leading manner, if one replication/reproduction is better than none, why aren't two better than one? This question, and others like it, imply that observed rates of RR attempts that are less than 100% may not be suboptimal in a broader sense.

The paper specifies “When we examined the 177 individual studies replicated in the 47 documents, we found only a minority of them referred to registered protocols”. We feel this accurately reflects the data; the term modest is not used. We have added to the discussion to consider the points raised: “We acknowledge, however, that mandates for registration are rare and exist only in particular disciplines and for specific study designs.”

On a different note, I am concerned about how the coding was done. Specifically, it appears that co-authors served as "screeners" of how to categorize certain articles and article attributes. In other words, they made decision that influenced the data that they were analyzing. The risk is that people who know the hypotheses and desired outcomes implicitly bring that knowledge to coding decisions. An established best practice in fields where this type of coding is done is to first train independent coders who are unaware of the researchers; hypotheses and then conduct rigorous inter-coder reliability assessments to document the extent to which the combination of coders and categorical framework produce coding outcomes that parallel the underlying conceptual framework. Such practices increase the likelihood that data generating processes are independent of hypothesis evaluation processes.

Piloting was indeed undertaken to train reviewers to ensure consistency. This has been further specified by adding: “Prior to extraction a series of iterative pilot tests were done on included documents to ensure consistency between extractors.”

For these reasons, I am very interested in the questions that the authors ask, and what they find, but I am not convinced that a number of the empirical claims pertaining to comparisons and magnitudes will generalize even to larger populations of articles in the stated disciplines.

We have expanded the discussion about generalizability by adding: “Collectively these study design decisions and practical challenges present limitations on the overall generalizability of the findings beyond our dataset.”

I would like to see the Discussion section rewritten to focus on what the findings, and methodological challenges, mean for future work. In the current version, there are a number of speculative claims and generalizations that do not follow from the empirical work in this paper. These facts and some variation in the focus of the text in that section make the discussion longer than it needs to be and may have the effect of diminishing the excellent work that was done in the earlier pages.

The reviewer has provided helpful discussion above which we have addressed as per our rebuttal notes. We feel the issues regarding potential concerns about generalizability are now stressed in the discussion.

Reviewer #3 (Recommendations for the authors):The authors provide a fairly clear and accurate summary of the current questions around reproducibility in the scientific literature. I was impressed that they thoughtfully included a description of the areas where their utilized methodology was different than what they had proposed in their protocol and explained the reasons for those differences. That kind of documentation is valuable and fosters transparency in the research process.The results of their analysis are provided in both narrative format and in tables and appendices. The authors effectively characterize the different, and sometimes difficult, decisions made in parsing the results of their search. While much attention has been paid to replication and reproducibility in recent years, the nature of the results reflects the reality that the replication studies themselves may have reporting issues, and it may not always be possible to ascertain how to link the results of such studies with the original work. The study results meaningfully add to the current, small body of literature examining the meta-scientific issues of reproducibility and replication of scientific findings.The paper is cautious in its approach to interpretation, appropriately using language such as "this may suggest that…" to better ensure that readers do not draw inappropriately firm conclusions. This is a descriptive paper, and so readers, like the authors, would best be served by limiting strong inferences based on the findings. At the same time, the authors helpfully suggest numerous meaningful pathways forward to advance reproducibility research in multiple scientific domains. Such studies would in turn facilitate more powerful inferences.In preparing this review, the substantive majority of recommendations that I had for the authors related to points of clarity conveyed in private comments rather than overt weaknesses or concerns.LL50-52: The meaning of this sentence is not entirely clear (though the paragraph can still be understood when skipping the sentence). Please consider rewriting to capture your point more closely.

Agree. We have deleted this line.

LL106-108: Since that analysis was published in 2014, and given the recent (albeit anecdotal) increased emphasis on reproducibility, it may be worth noting the date of the study in the text itself (or the termination date of the search used for the 0.13% statistic).

We have specified 2014 now.

L126: In this context, "some effect in inbreeding" is unclear.

We have reworded to “some effect of researcher familiarity”.

LL135-136: There may be value in disentangling the link between reproducibility and harms in the Introduction more explicitly. In the cited study (Le Noury et al), medications were utilized in a trial where reanalysis indicated that they were unlikely to produce a clinical benefit and each resulted in plausibly increased risk of harm. But it was not the failure to replicate, per se, that led to possible harms. Rather, the concern was clinical decision making predicated on the results of the original study, the results of which "stood" pending reanalysis. This may seem like a pedantic point but I think it speaks to the role not only of the original study itself, but also the degree to which decisions were made based on the findings of a single study.

Great point to consider. We have amended this to read: “In medicine, studies that do not reproduce in clinic may exaggerate patient benefits and harms^14^ especially when clinical decisions are based on a single study.”

L240: Can you provide a short clarification of what you mean by calibration test – such as inserting a parenthetical example "…performed a calibration test (e.g., did X) prior to…"

We have specified: “Specifically, a series of included documents were then extracted independently. The team then met to discuss differences in extraction between team members and challenges encountered before extracting from subsequent documents. This was repeated until consensus was reached.”

Table 1: The box indicating a cross section of number of authors and all studies appears to have inaccurate/incomplete information.

Thanks for catching this. This has been updated.

Table 1: The way the data are structured is a little unclear. For example, for "All Studies" the "discipline" row uses a denominator of total replication studies, whereas "year of publication" in the same column uses a denominator of the number of papers. While I see why this was done, I think additional clarity or uniformity in how the data are presented would be helpful.

This distinction has been reflected in the footnotes.

LL300-301: Could you briefly indicate whether you think that excluding documents unavailable in full via your library may have affected the results (e.g., was this 11 quantitative replications, as assessed by abstracts? or was it a mixed bag?).

We have added this to the limitation section by specifying: “We also were not able to locate the full-text of all included documents which may have impacted the results.”

L384: Does the count for psychology include the three dozen or so replication studies that were unusable for this paper? If so, do the authors think that their inclusion here serves to accurately represent the replication efforts in psychology or may artificially inflate the frequency? I don't have the perspective to know which is the case, but I think the question is worth considering.

It is a good point, but hard to know without accessing the full-text. In addition to the line above, we now specify: “The impact of these missing texts may not have been equal across disciplines.”

LL397-398: Would you be willing to share any information (if it exists) about whether independent and separate replications differ from those embedded within experimental protocols in meaningful ways (not just in terms of the obvious process differences)?

We did not examine this.

L417: Are you referring to registered protocols for the replication studies? Or registered protocols for the original study that can be used in completing the replication?

We refer to registered protocol of the replication studies. This has not been further clarified by modifying the line to read “we recorded whether a registered protocol for the replication study was used”.

LL430-432: It might be helpful for readers to be briefly informed about what "the current environment" means in practice, as there is a lot of possible variability. For example, I might assume that there is an overemphasis on novelty at the desk editing stage, but it's not clear whether the authors were even thinking of that in particular.

That’s correct. We have modified this line to read: “Of note, when no new data are generated, it may be difficult in the current research environment, which tends to favor novelty, to publish a re-analysis of existing data that shows the exact same result^31,32^.”

LL434-448: There is a lot of interesting discussion in this paragraph that gets somewhat "jumbled together" in this paragraph. First, I'm not sure that it makes sense to characterize the finding as biased in L436. Rather, you found that for replication studies that were not part of the same paper, lack of overlap was common, and you do not make any assertions about studies where that was not the case (but presume the overlap might be higher – and in fact would by definition be nearly 100% since authors do not change partway through papers). Second, the larger question about what constitutes a replication is related to, but seemingly separate from, the initial discussion. Even if the authors agree on what a replication would look like for a specific study, it may not be coherent in the context of the broader field's common understanding of reproducibility.

We have re-read this section to address the jumble and streamline.

[Editors’ note: what follows is the authors’ response to the second round of review.]

The manuscript has been improved but there are some modest remaining issues that need to be addressed, as outlined below:

Reviewer #3 (Recommendations for the authors):I would like to thank the authors for their response to my original comments. In most cases, I found that the authors' revisions were responsive to my concerns. In the instances where I did not find that to be entirely the case, I have noted it below using the new line numbers.L126: Thank you for revising the line about inbreeding. However, I think that the replacement phrase is still a little unclear. The sentence is structured as an 'either/or' and the first part suggests familiarity with the study method may increase likelihood of reproducing the same findings. So I am not sure how "researcher familiarity" functions as the alternative. The sense I get is that the authors may be hedging around suggesting that in some cases, the author's approach to research (perhaps unusually rigorous, or unusually sloppy) may carry over between studies and explain some of the variability in findings. However, that is an assumption. In any case, the authors might benefit from being very direct about their theory here.

We have amended the line to read: “This may suggest that detailed familiarity with the original study method increases the likelihood of reproducing the research findings.”

L110: The phrase "…and some disciplines do worse in this regard" might be interpreted to have two meanings. I interpret it to mean that some disciplines have fewer studies where reproductions have been attempted. However, "do worse" might also be taken to mean that some disciplines' studies are less likely to successfully have their results reproduced, which is a different concept entirely.

We have amended this line to read: “Most scientific studies are never formally reproduced and some disciplines have lower rates of reproducibility attempts than others.”

LL404-405: I appreciate this revision made in response to reviewer #1. I am unsure about whether an assertion of "unique cultures" is significantly different than the original assertion of which cultures have it as a normative value. I think the line may be fine to retain, but should be constrained with a caveat e.g., "This may suggest unique cultures around reproducibility across different disciplines, but further study is needed to determine whether this is truly the case, and our study should not be taken as proof of differences between disciplines."

We have amended this line to read: “This may suggest unique cultures around reproducibility in distinct disciplines, future research is needed to determine if such differences truly exist given the limitations of our search and approach.”